# A Short Overview of Changes in Inflammatory Cytokines and Oxidative Stress in Response to Physical Activity and Antioxidant Supplementation

**DOI:** 10.3390/antiox9090886

**Published:** 2020-09-18

**Authors:** Shima Taherkhani, Katsuhiko Suzuki, Lindy Castell

**Affiliations:** 1Department of Exercise Physiology, Faculty of Sport Sciences, University of Guilan, Rasht 4199843653, Iran; shimataherkhani@msc.guilan.ac.ir; 2Faculty of Sport Sciences, Waseda University, 2-579-15 Mikajima, Tokorozawa 359-1192, Japan; 3Green Templeton College, University of Oxford, Oxford OX2 6HG, UK

**Keywords:** exercise, cytokine, oxidative stress, muscle damage, reactive oxygen species (ROS), nuclear factor-κB (NF-κB), antioxidants

## Abstract

Excessive release of inflammatory cytokines and oxidative stress (OS) are triggering factors in the onset of chronic diseases. One of the factors that can ensure health in humans is regular physical activity. This type of activity can enhance immune function and dramatically prevent the spread of the cytokine response and OS. However, if physical activity is done intensely at irregular intervals, it is not only unhealthy but can also lead to muscle damage, OS, and inflammation. In this review, the response of cytokines and OS to exercise is described. In addition, it is focused predominantly on the role of reactive oxygen and nitrogen species (RONS) generated from muscle metabolism and damage during exercise and on the modulatory effects of antioxidant supplements. Furthermore, the influence of factors such as age, sex, and type of exercise protocol (volume, duration, and intensity of training) is analyzed. The effect of antioxidant supplements on improving OS and inflammatory cytokines is somewhat ambiguous. More research is needed to understand this issue, considering in greater detail factors such as level of training, health status, age, sex, disease, and type of exercise protocol.

## 1. Introduction

Cytokines are small soluble proteins that are capable of altering the demeanor of other cells locally which can regulate inflammatory and immune responses [1,2,3,4]. In general, inflammatory cytokines are divided into two central types, pro-inflammatory cytokines (tumor necrosis factor alpha (TNF-α) and interleukin (IL)-1) and anti-inflammatory cytokines (IL-1 receptor antagonist (IL-1ra), IL-4, IL-10, IL-11, and IL-13) [5,6]. According to the Scheller et al. [7] study, the IL-6 cytokine has both pro- and anti-inflammatory properties. Cytokines have specific functions within the body, such as controlling, initiating, and modulating inflammatory responses [8]. On the other hand, their function depends on the extent to which they affect immune cells and leukocytes [9].

Regular physical activity of light to moderate intensity supplies a multitude of health benefits, including decrements in the risk of various diseases such as cardiovascular disease, diabetes, cancer, dementia and Alzheimer’s disease [10], adverse blood lipid profile [11], and improved non-alcoholic fatty liver disease (NAFLD) [12] and disabling non-motor symptom in Parkinson’s disease (PD) [13]. In addition, these types of activities relieve antioxidant defense incrementally and diminish oxidative stress (OS), which ultimately leads to a reduction in mortality [9,13,14,15]. It should be noted that free radicals and reactive oxygen species (ROS) were discovered in 1970 [16]. These risk factors induce cellular dysfunction, redox imbalance, and OS [17]. Macromolecules such as lipids, nucleic acids, and proteins are damaged during strenuous physical activity. Oxidation of peripheral chains of amino acids and disintegration of polypeptides, degradation of unsaturated fatty acids and phospholipids and mutations, respectively, are among the side effects of protein, fat, and deoxyribonucleic acid (DNA) damage [18]. With the onset of physical activity, anaerobic and aerobic metabolism and the need for energy by active muscles increase [19].

Alternatively, consumption of antioxidant supplements by different individuals can be a reasonable strategy to reduce OS and enhance recovery and performance. There are impressive benefits for these supplements, but different studies have demonstrated their destructive effects on responses of skeletal muscle to exercise. These impairments resulted in adaptive changes within skeletal muscle such as an attenuation of normal redox-signaling pathways [20].

The purpose of this review is to organize research that has investigated changes in inflammatory cytokines and OS in response to physical activity and antioxidant supplementation. The first part of this review describes the response of cytokines to exercise in four continuous sections, including the origin of cytokines, the effect of cytokines on the body, the mechanism of action of cytokines, and the factors influencing the production of cytokines. Next, we looked at the OS response to exercise in six small areas, including the impact of strenuous exercise on oxidative stress and the effects of high levels of ROS, the main causes of oxidative stress during and after exercise, the source of ROS production, the various types of ROS, old and new views on ROS, and the positive effect of ROS at low levels in the body during exercise. In the next section, the measurement of ROS in the body is briefly discussed. This also includes the oxidation-reduction (redox) control of cytokine composition in three separate sections called the redox-sensitive nuclear factor-κB (NF-κB) signaling pathway, calcineurin-NFAT signaling pathway, and redox-sensitive heat shock proteins (HSPs) signaling pathway. The next two sections discuss the effects of antioxidant supplements on inflammatory cytokines and oxidative stress. In addition, in the last section, antioxidant supplements as a pro-oxidant are briefly discussed.

## 2. Cytokines Response to Physical Activity

Cells such as fibroblasts, endothelial cells, neutrophils, activated macrophages, and damaged muscle cells can secrete protein molecules, for instance, cytokines. In addition, muscles can generate these cytokines when the motor units contract. After secretion from different cells, these cytokines exert effects such as proliferation, differentiation, and survival of cells in the body, especially immune cells [21]. Excessive release of cytokines is a triggering factor in the onset of chronic diseases [22]. Regular physical activity can augment immune function and dramatically barricade the spread of these cytokines and their irredeemable damage [23]. Diverse mechanisms for reducing inflammatory cytokines have been defined by regular physical activity as described below.

One of the most effective of these mechanisms is the role of physical activity in reducing the production of pro-inflammatory cytokines due to strengthening the immune system. Other mechanisms that can affect adipose tissue include reducing inflammation in the tissue and finally improving hypoxia. In addition, skeletal muscle tissue with the secretion of anti-inflammatory myokines has a superior effect on reducing the cytokines of the muscles during physical activity. Myokines are a type of cytokine that have endocrine and paracrine effects and are mainly secreted by muscle fibers [24]. Endothelial cells also help reduce these cytokines by reducing the adhesion of leukocytes [25].

Exhaustive physical activity can adversely affect the immune system by increased expression of glycoprotein and myosin levels [26]. This is especially important for athletes, because even minor infections can have serious consequences, such as decreases in physical activity performance and training [27,28]. The secretion of these cytokines is regulated by muscle glycogen breakdown and Ca^2+^ [28].

The results of studies that have examined the increase in post-exercise inflammatory cytokines are contradictory. One study found that, if subjects exercised at 60 to 65% of maximal oxygen intake (VO_2max_) for 180 min, their plasma concentration of pro-inflammatory cytokines remained high for up to 24 h after exercise [29]. Another study demonstrated that pro-inflammatory cytokines also remained high for up to 1 h after a marathon race [8]. However, this has not been confirmed in other studies in which it was concluded that there was no increase in the plasma concentration of these cytokines after marathon running [2,3,30].

One study has also shown that the rate of change in inflammatory cytokines after strenuous physical activity depends on a person’s weight. In obese people, the increase in cytokines (IL-6 and IL-10) is estimated to be higher than in lean people [31]. In another study, the researchers also looked at the rate of change in these cytokines after 9 months of resistance training. They concluded that IL-6 decreased after the end of the training period [32]. Age is one of the factors that can be of special importance in creating and expanding inflammatory indicators. Various studies have shown that resistance training is able to eliminate low-grade inflammation in the elderly [33]. The formation of excess aldehyde-protein adducts (in particular, 4-hydroxynonenal (HNE) protein excess compounds and malondialdehyde (MDA)) is one of the reasons for the increase in oxidative stress with age [34]. Today, interest in examining the different effects of combination of endurance and resistance training on various aspects of body health has increased exponentially among researchers. One study found that combination exercises were more forceful at reducing inflammatory cytokines than performing each exercise alone [35].

Altogether, the results of studies that have examined the effect of exercise on pro-inflammatory cytokines are contradictory. The cause of this disagreement may be the presence of several factors such as level of training, health status, age, sex, disease, and type of exercise protocol (volume, duration, and intensity of training) [26,33]. 

## 3. Oxidative Stress Responses to Physical Activity

One of the factors that ensures good health is regular physical activity. However, if physical activity is done intensely and irregularly, it is not only unhealthy but can also lead to muscle damage, OS, and inflammation in the body. ROS can be produced independently of acute or chronic physical activity [16]. Of course, factors such as increased mitochondrial activity, ischemia/reperfusion, and NAD(P)H oxidase complex/leukocyte activation during and after physical activity are the main causes of ROS production [36]. In fact, OS and ROS accumulation are earnest consequences of repetitive contractions in skeletal muscle [37]. It is well documented that the production and accumulation of ROS occur in the mitochondria and during the electron transfer process, and that these molecules include the superoxide anion (O_2_·^−^), hydrogen peroxide (H_2_O_2_), and hydroxyl radicals (^·^OH). On the other hand, the aerobic production of ATP in mitochondria occurs via the process of oxidative phosphorylation [36,37,38]. Until a few years ago, ROS was thought to be one of the most damaging factors in muscle during and after exercise, but recent studies have shown that ROS can regulate muscle signals to exercise, as well as various oxidative phenotypes in muscle, by regulating different signals [36,39,40,41,42]. In fact, exercising with low intensity and duration produces low to moderate levels of ROS [36]. ROS production during exercise participates in various processes such as increased protein synthesis, activation of insulin signaling, mitochondrial biogenesis, activating signaling pathways, controlling muscle production, gene expression, and positive regulation of antioxidants. Studies have even shown that eliminating ROS and disrupting its production regulation may increase the risk of diseases such as obesity and type 2 diabetes [43]. However, these effects may be reversed under conditions such as increased duration and severity of exercise activity, in which case, not only does this not support the cells, but it can also lead to cell damage [2,36]. Free radicals can improve muscle adaptation to exercise. Studies have also shown that over-accumulation of ROS, followed by decreased antioxidant capacity, has serious consequences such as damage of protein, DNA, and lipid, together with the activation of inflammatory pathways [41,44].

## 4. Measuring ROS in the Body

There are plenty of procedures to measure ROS in vivo non-invasively. However, recently researchers used L-012 in their experiments to evaluate these molecules by optical imaging (OI). L-012 is a luminol-based chemiluminescent (CL) excavator commonly used to measure NADPH (Nox)-derived superoxide (O_2_^−^) [45] and other ROS [2]. Of course, one of the actions and reactions between gene expression and immune cell migration is temporal dynamics that is related to ROS production and its interplay with inflammatory pathways, particularly, NF-κB [45]. There is very little information about temporal dynamics: however, one way to control ROS and even NF-κB activity in the body in a non-invasive way is to determine temporal dynamics, which helps therapeutic interventions, but it has a short lifespan of msec-min, and thus the great variety of ROS in the body faces challenges [46].

## 5. Oxidation-Reduction (Redox) Control of Cytokine Composition

Maintaining the balance of cellular oxidant-antioxidants in the body of mammals is established by several signaling pathways. In this section, three important pathways sensitive to oxidative stress are described.

### 5.1. Redox-Sensitive NF-κB Signaling Pathway

The recognition of NF-κB transcription factors by Baltimore and Sen dates back to 1986 [47]. NF-κB usually acts like a double-edged sword. These proteins contain homo- and heterodimers such as c-Rel (REL), RelA/p65 (RELA), RelB (RELB), p50/p105 (NFKB1), and p52/p100 (NFKB2) have the ability to maintain cell survival through the expression of NF-κB target genes [48,49]. They also play an important role in muscle protein turnover and regulating the body’s anti-inflammatory and immune responses, and are one of the most important signaling pathways for maintaining cellular oxidant–antioxidant balance known. This pathway is activated by ROS [49,50,51]. In fact, ROS can activate the NF-κB signaling pathway sensitive to redox changes, and then lead to increased expression of the gene for antioxidant enzymes in stressed oxidative tissues. NF-κB is also involved in cell death [52,53].

ROS can somewhat modulate NF-κB responses in causing cell death and can lead to necrotic and apoptotic cell death. It should also be noted that ROS can be reduced through NF-κB target genes, thus helping to maintain cell survival [54]. NF-κB function is primarily controlled by a set of proteins named IκBs [55]. In general, these proteins are divided into two types, typical or classical (IκBα, IκBβ, IκBε, and IκBδ (p100) and atypical (IκBζ, Bcl-3, IκBNS, IκBη, and IκBL), depending on their location in the cytoplasm or cell nucleus, respectively [56]. One of the common features of these two types of proteins is the presence of an ankyrin repeat domain (ARD), which itself consists of six to eight single ankyrin repeats (ANK). In addition, these two types of proteins show different behaviors after stimulation with NF-κB-inducing factors. Under such conditions, nuclear IκBs levels increase and cytoplasmic IκBs levels decrease [57]. In fact, typical proteins can inhibit the NF-κB by retaining NF-κB-dimers within the cytoplasm. These proteins bind to NF-κB-dimers due to their nuclear export signal sequence (NES), thus masking the nucleus binding sites by nuclear localization signal sequence (NLS) of Rel proteins, leading to NF-κB-inhibition. On the other hand, atypical proteins bind to Rel family proteins in the nucleus and modulate transcriptional activity of NF-κB (Table 1). These proteins are also able to regulate the maturation, differentiation, and activation of immune cells [47,58,59].

One study has shown that the NF-κB signaling pathway is activated by performing acute and exhaustive exercise in rat skeletal muscles [60]. In fact, the regulation of various cellular functions and the differentiation of gene clusters are associated with the activation of the NF-κB pathway by exercise [61]. Since the cytoplasm is where NF-κB proteins are located, exogenous antioxidants must pass through the cell membrane to exert their effects inside the cell [62]. These antioxidants have anti-diabetic and anti-inflammatory properties. By activating various anti-inflammatory pathways, they can prevent the activity of anti-inflammatory cytokines from continuing [63]. In addition, it has recently been investigated that a variety of stimulants such as phorbol ester, lipopolysaccharide, cycloheximide, cytokines IL-1 and TNF-α, and even H_2_O_2_, which can activate the NF-κB transcription factor, are inhibited by antioxidant supplements [64]. One study has concluded that these supplements prevented the induction of inflammatory cytokines, especially TNF-α. This is done by NF-κB signaling pathways using lipopolysaccharide molecules [60]. Another study has shown that NF-κB activity may be inhibited in the duodenal mucosa of pigs by consumption of grape seed and grape marc extract (GSGME) supplementation [63]. In addition, another study has mentioned that some antioxidant supplements such as vitamin E were able to protect the body’s membranes against lipid peroxidation [65]. The NF-κB signaling pathway may be inactivated by taking vitamin E in both in vivo and in vitro models. Blocking this pathway depends a lot on reducing OS [66].

### 5.2. Calcineurin-NFAT Signaling Pathway

Another signaling pathway to maintain cellular oxidant–antioxidant balance is the calcineurin-NFAT signaling pathway, which can be activated by ROS during exercise [67]. In general, mitochondrial respiration is the site of ROS production, which exhibits different behaviors at low and high levels. Thus, this molecule enhances NFAT activity at low levels and inhibits it at high levels [68]. The calcineurin-NFAT pathway regulates gene expression in response to changes in intracellular calcium concentration and plays important roles in many tissues, including lymphocytes, neurons, and myocytes [69]. Calcineurin is the major soluble calmodulin-binding protein in brain extracts. Calcineurin is one of the protein phosphatases that can be regulated by calmodulin and calcium. Initially, Ca-microdomains activate calcineurin. Then, with dephosphorylation of the NFAT isoforms, they are transferred into the cell nucleus. NFAT is also present in skeletal muscle as one of the sensors sensitive to the activity of the nervous system [65,66]. This sensor is also important in various functions such as producing and activating lymphocytes and differentiating myocardial cells. Calcineurin binds to calmodulin and inhibits the calmodulin stimulation of cyclic nucleotide phosphodiesterase. In addition, calcineurin inactivates an enzyme that is regulated by calmodulin. In general, NFAT (transcription agents) has two members, including calcium-regulated isoforms and tonicity [70,71,72]. The NFAT family consists of 4 calcium activated members (NFATc1–NFATc4) and one form which senses osmotic stress (NFAT5). In general, peripheral lymphocytes are the site of expression of NFAT1,2,4 transcription factors. Of course, the NFAT4 transcription factor is expressed specifically in thymocytes and the NFAT1 and NFAT2 transcription factors are expressed in peripheral T cells [73,74].

### 5.3. Redox-Sensitive Heat Shock Proteins (HSPs) Signaling Pathway

Eukaryotic and prokaryotic cells can produce heat shock proteins (HSPs) in response to different insults [71]. These proteins were discovered in 1962 by Ritossa [75], and may be present both inside the cell and on the cell membrane [76]. In general, HSPs exhibit different behaviors in different situations, and so some of these proteins can be produced constructively and effectively under natural conditions. However, when these proteins are overproduced under stressful conditions such as ischemia, exercise, oxidative stress, heat, hypoxia, high temperature, inflammation, free radicals and ROS, and acute or chronic diseases, they lead to denaturation of proteins. On the one hand, immune reactions and tumor formation play an important role in the development of OS associated with aging [75,76,77]. On the other hand, in such stressful situations, HSPs may protect the body’s cells through various processes such as the initiation of protein folding, repair, refolding of misfolded peptides, and possible degradation of irreparable proteins. In fact, various studies have shown that free radicals have destructive effects at different cellular levels. However, different organs of the body can greatly counteract free radicals and ROS by increasing HSPs expression. In fact, these proteins exert their protective effects due to their role in repair, polypeptide folding, the degradation of irreparable peptides, assembling and translocation of organelles across membranes [78]. Factors such as molecular weight, amino acid composition, and various cellular functions have caused HSPs to be generally divided into two groups with high and low molecular weight. High molecular weight HSPs are usually dependent on ATP and usually weigh between 60 and 110 kDa. These types of proteins, using ATP-dependent allosteric control, lead to the binding and folding of emerging proteins. On the other hand, other types of HSPs tend to have a small molecular weight between 15 and 43 kDa. These are known as heat shock protein β (HSPβs). Unlike high molecular weight HSPs, these proteins are not dependent on ATP, and usually play a key role in the embryo development process [75,79]. HSPs play an important role in facilitating various processes such as protection from denaturation and protein folding [72]. The link between adaptive and innate immune systems is another function of these proteins in the body [80]. They are also one of the most important indicators of stress and cell damage [81]. When the body is exposed to environmental pressures such as hypoxia and oxidative damage, antioxidant compounds can help to re-establish cellular homeostasis and prevent excessive increase of HSPs [82,83,84]. One of the positive effects of HSP70 is the elimination of genetic toxicity caused by ROS, which has led to DNA fragmentation [75]. With increasing stimulation of cell membrane lipid biolayer, the threshold of induction of HSP70 expression is usually reduced. Insufficient or excessive antioxidants are one of the important factors that can modulate the activity and subsequent synthesis of HSPs. In fact, antioxidants known as ROS neutralizers are able to inhibit HSPs genes [76].

## 6. The Effect of Antioxidant Supplements on Inflammatory Cytokines

Some of the most prominent and fundamental roles of cytokines, especially IL-6 during and after exercise, are their anti-inflammatory responses to exercise [85]. Various mechanisms, including ischemia/reperfusion, electron leakage at the mitochondrial electron transport chain, and activation of endothelial xanthine oxidase, inflammatory cell response, and autooxidation of catecholamines, are involved in the production and spread of inflammatory cytokines [85,86,87]. Two studies have shown that increased plasma IL-6 concentrations during exercise can be lowered by antioxidant supplements [88,89]. Of course, the effect of these types of supplements on changing the concentration of cytokines is ambiguous and no definite conclusion can be drawn about it. For this reason, there is still a need to do more research on this issue. Some studies have reported reduced neutrophil function after endurance training (2.5 h at 60% VO_2max_); the function did not improve even with vitamin C intake (in a 2-week period with a dose of 1000 mg/day) [90]. In another study, Aguiló et al. [91] concluded that voluntary male recreational well-trained athletes consuming vitamin C (500 mg/day) for 15 days showed no reduction in anti-inflammatory cytokines such as IL-6 and IL-10 which normally increase after endurance exercise (15-km running).

However, the effect of concomitant use of antioxidant supplements is one of the topics to which researchers have paid special attention in recent years. In one article, Vassilakopoulos et al. [92] examined the effects of taking several antioxidant supplements simultaneously. Healthy individuals participated in the study and were given a combination of supplements such as allopurinol 600 mg/day for 15 days, vitamin A 50,000 IU, vitamin C 1000 mg and vitamin E 200 mg/day for 60 days, and N-acetylcysteine 2 g/day for 3 days. The changes in plasma cytokine levels (IL-1β, IL-6, and TNF-α) were measured after two resistive breathing sessions at 75% of maximum inspiratory pressure. The authors concluded that OS was primarily responsible for increased respiratory resistance due to inflammatory cytokines. McAnulty et al. [93] also tested the effect of concomitant use of resveratrol and quercetin supplements on healthy trained male adults. They consumed 225 mg quercetin and 120 mg resveratrol for 6 days and 450 mg quercetin and 240 mg resveratrol on day 7 just prior to exercise. Their measurement variables included oxygen radical absorptive capacity (ORAC), ferric-reducing ability of plasma (FRAP) and Trolox equivalent antioxidant capacity (TEAC), protein carbonyls and F_2_-isoprostanes and C-reactive protein (CRP), and the cytokine IL-8 in order to determine the rate of change in antioxidant capacity, OS and inflammation, respectively. It was finally concluded that resveratrol and quercetin supplements significantly reduced exercise-induced lipid peroxidation. However, it did not alter plasma antioxidant status and inflammation.

In another study, Bailey et al. [94] gave their subjects (healthy young men) a combination of antioxidant supplements (vitamin C and E). These supplements included 400 mg vitamin C (ascorbic acid), 268 mg vitamin E (RRR-α-tocopherol), 2 mg vitamin B6 (pyridoxine hydrochloride), 200 µg vitamin B9 (folic acid), 5 µg zinc sulphate monohydrate, and 1 µg vitamin B12 (cyanocobalamin), which they took twice a day for 6 weeks. After completing the period of supplementation, the subjects performed shuttle running for 90 min. The researchers concluded that inflammation and OS caused by exercise were not reduced by taking the supplements. In addition, the supplements had no effect on improving post-workout recovery due to muscle damage. In ultramarathon sports (long-duration, high-intensity activity), most of the body’s physiological systems are affected [95]. Therefore, in this type of activity, in order to supply oxygen to the muscles involved, the amount of blood flow to these muscles must be increased [96]. However, the consequences of this type of exercise include hemostasis disturbance, metabolic stress, and adverse physiological responses [95]. The reasons include inadequate recovery, increased fatigue, accumulation of stress hormones such as cortisol, defects in the immune system, increased inflammatory cytokines, and impaired iron status [97].

In one study, ultramarathon athletes were given quercetin as an antioxidant supplement for three weeks at a dose of 1000 mg/day. After three weeks, the athletes competed in a 160-km Western States Endurance Run (WSER) race. The results of this study included enhancing quercetin plasma levels, but the supplementation failed to relieve muscle damage and inflammatory cytokines [98]. In another study, Tongtako et al. [99] tested the effectiveness of exercise with vitamin C at a dose of 2000 mg/day for 8 weeks in patients with allergic rhinitis, who had to undertake treadmill walking at 65–70% heart rate reserve for 30 min/session, three times a week during the eight weeks. They reported that the cytokine profiles such as IL-4 of these patients were significantly reduced. Numerous studies have proven the effectiveness of various antioxidant supplements (such as Phlebodium decamanum) in increasing antioxidant defenses and simultaneously reducing oxidative stress indicators such as isoprostane and 8-OHdG following ultraendurance activity. These types of supplements are able to create a strong defense against pro-inflammatory cytokines such as TNF-α. Additionally, the effectiveness of these supplements does not necessarily depend on a person’s level of physical activity and can be effective for both professional and amateur athletes [100,101]. Diaz-Castro et al. [102] confirmed this in a study in which they supplemented their subjects with co-enzyme Q10 (a lipid-soluble antioxidant) before undertaking strenuous exercise (ultraendurance). The results of this study included reduction of muscle damage, oxidative stress, and inflammatory cytokines. In addition, Ochoa et al. [103] obtained the same positive results associated with antioxidant supplements, although they attributed this to the fact that they supplemented their subjects with melatonin. It is worth noting that Gomez-Cabrera et al. [16] advised against supplementing with the antioxidant vitamin C because it hampered training-induced adaptations (see also B. Ekblom in Castell et al. [104] who similarly does not recommend co-enzyme Q10 because of blunting adaptations).

Among the types of muscle contractions, it has been proven that, during eccentric (ECC) actions, more force per unit of muscle size is produced than in the other two concentric (CON) or isometric (ISOM) actions. Other features that distinguish ECC include less metabolic need and less activation of the number of motor units/time. In addition, ECC causes more hypertrophy [105]. On the other hand, ECC has adverse consequences such as inflammation, exercise-induced muscle damage (EIMD), reactive oxygen nitrogen species (RONS) production, excessive fatigue, and impaired body function [106,107,108]. Additionally, delayed-onset muscle soreness (DOMS) is another side effect of this type of exercise [109]. Sports scientists have attributed the damage to the body to the migration and mobilization of neutrophils and macrophages [110]. These researchers have come up with various solutions to reduce the effects of ECC. In one study, subjects took a beetroot juice (rich in antioxidants) for 3 days after ECC, and muscle pain caused by exercise significantly reduced, whereas countermovement jumps performance dramatically dropped [111]. In another study, Koenig et al. [112] supplemented their subjects with avenanthramides (AVA) antioxidant supplement. The results of this study showed that this supplement was able to reduce inflammatory cytokines caused by ECC (Table 2).

## 7. The Effect of Different Doses of Antioxidant Supplements on OS

The effect of taking antioxidant supplements with a single dose has been documented in several studies, but the effect of taking these supplements in different doses is ambiguous. In one study, male Wistar rats were given Radix Pseudostellariae polysaccharides (RPPs) antioxidant supplement daily for 4 weeks in different doses (100, 200, and 400 mg/kg body weight). After supplementation, the rats had to swim to the point of exhaustion. The results showed that high-dose antioxidant supplementation was more effective in improving the levels of antioxidant enzymes such as SOD, GP, X and CAT. At the same time, there was a further reduction in OS indicators [113]. In addition, in another study, healthy mice were given an antioxidant compound named grass carp protein or peptide with low (1 mg/g.d) and high (5 mg/g.d) doses every day before swimming exercise for 28 days. Consumption of the high-dose compound significantly increased the levels of antioxidant enzymes such as SOD and CAT [114,115].

Zheng et al. [116] gave rats low (20 mg/kg/d) and high (100 mg/kg/d) dose wheat peptide antioxidant supplement together with incremental swimming exercise for 4 weeks. The researchers eventually concluded that taking both doses of the supplement could boost GPX antioxidant levels. However, fatigue was delayed by taking a high-dose supplement compared with a low dose. On the other hand, Liu et al. [117] showed that if male Sprague-Dawley rats were given a antioxidant supplement of lycopene (a relatively high dose (7.8 mg/kg/d) and a relatively low dose (2.6 mg/kg/d) for 30 days), there was no significant difference between the use of different antioxidant doses in the enhancement of glutathione antioxidant enzymes. Additionally, in a recent meta-analysis of 50 studies, it was clearly stated that exercise-induced muscle pain did not decrease even with high-dose antioxidant supplements [118] (Table 3).

## 8. Antioxidant Supplements as a Pro-Oxidant

Although the positive and effective effect of antioxidant supplements on the prevention of OS and even the treatment of various diseases has been proven to some extent, there are still studies that have pointed out the harmful, adverse, and pro-oxidant effects of these supplements. For this reason, these supplements should be prescribed with extreme caution and by an experienced nutritionist for different individuals over a period of time [119].

In one of the most authoritative published articles, researchers compared damage and OS caused by performing an eccentric exercise and consumption of N-acetylcysteine (NAC) (10 mg/kg b.w) and vitamin C (12.5 mg/kg b.w) at the same time for 7 days. They concluded that the rate of oxidative stress increase after supplementation was much greater than the harm caused by eccentric exercise alone. One of the reasons why these supplements show a pro-oxidant property both in vitro and in vivo is their high reactivity with some intermediate metals such as iron [120]. In another study, male mice were fed 200 mg/kg b.w of the antioxidant supplement named acacia polyphenol and ran on a treadmill for 10 min at a speed of 15 m/min. It was concluded that, although this supplement was able to reduce OS in skeletal muscle, it increased OS in the liver [121].

To counteract the pro-oxidant effects of antioxidant supplements, the body’s antioxidant defense system must be strengthened. In fact, it is well established that antioxidant enzymes can greatly relieve RONS. To this end, RONS-related pathways and mechanisms must be regulated for gene expression of antioxidant enzymes and cellular response to oxidative stress [122]. One of the important pathways in this field is Nrf2-keap1 (nuclear factor erythroid 2-related factor 2- Kelch-like erythroid cell-derived protein with cap ‘n’ collar homology-associated protein 1) pathway. This pathway plays a major role in the expression of genes associated with maintaining cellular homeostasis and redox status, counteracting oxidizing molecules, and controlling cell functions such as apoptosis, differentiation, and cell proliferation [123,124]. In fact, under oxidative stress, Nrf2 within the nucleus interacts with antioxidant responsive element (ARE), leading to cell survival. However, under hemostatic conditions, Nrf2 binds to the suppressor of the keap1 protein in the cytoplasm and is thus targeted by ubiquinine-dependent degradation in the proteasome [125,126]. On the other hand, many studies have recently concluded that Nrf2 can play an effective role in the prevention and treatment of diseases including exercise-induced organ damage and inflammation [122,123,126].

## 9. Conclusions

The production and accumulation of ROS and cytokines in the body act as a double-edged sword. In this way, they are able to cause oxidative damage and can also play an important role in activating signaling processes to create various adaptations in the body, such as increasing protein synthesis, activating insulin signaling, mitochondrial biogenesis, and positive regulation of antioxidants. On the other hand, when skeletal muscles contract, these muscles can secrete cytokines. Of course, factors such as muscle glycogen breakdown and Ca^2+^ can control the production of these molecules. Cytokines exert effects such as proliferation, differentiation, and survival of cells in the body. Many studies have investigated the fact that skeletal muscle contractility may be impaired by the use of exogenous antioxidants. However, low-level ROS molecules can partially facilitate the contraction of these muscles. In addition, the effect of antioxidant supplements on improving oxidative stress and inflammatory cytokines is somewhat ambiguous. More research is needed to understand this issue, taking into account factors such as the level of training, health status, age, sex, disease, and type of exercise protocol, in particular the volume, duration, and intensity of training.

## Figures and Tables

**Table 1 antioxidants-09-00886-t001:** The IκB family. Nuclear export signal sequence (NES), ankyrin repeat domain (ANK), nuclear localization signal sequence (NLS), and transactivation domain (TAD) are shown as blue, red, orange, and green ovals, respectively.

Typical	Atypical
IκBα 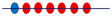	IκBζ 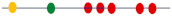
IκBβ 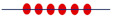	Bcl-3 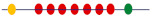
IκBε 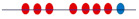	IκBNS 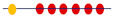
IκBδ (p100) 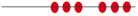	IκBη 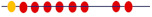
IκBL 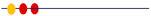

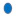
 NES 
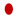
 ANK 
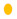
 NLS 
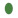
 TAD.

**Table 2 antioxidants-09-00886-t002:** The effect of antioxidant supplements on inflammatory cytokines.

Reference	Subjects	Exercise	Design	Supplement	Parameters	Results
**Davison and Gleeson** [90]	Healthy endurance trained males	2.5 h at 60% VO_2_max	Crossover (*n* = 9)	Placebo or Vitamin C (1000 mg day^-1^)	Cortisol	No positive effect on neutrophil function
					Adrenocorticotrophic Hormone	
					Interleukin-6	
					Oxidative stress	
					Neutrophil	
**Aguiló et al.** [91]	voluntary male recreational well-trained athletes	15-km run competition	A double-blinded study (*n*= 31)	Placebo or Vitamin C (500 mg day^-1^)	Vitamin A	No positive effect on reducing the inflammatory cytokines
					α-tocopherol	
					Malondialdehyde (MDA)	
					Cortisol	
					Creatine kinase	
					Aspartate aminotransferase	
					Lipid hydroperoxide	
					Vitamin C	
					IL-6	
					IL-10	
**Vassilakopoulos et al.** [92]	Healthy males	two resistive breathing sessions at 75% of maximum inspiratory pressure	Crossover (*n* = 6)	Allopurinol 600 mg/day	IL-1β	OS is responsible for increased respiratory resistance due to inflammatory cytokines
				Vitamin A 50,000 IU, vitamin C 1000 mg and vitamin E 200 mg per day	IL-6	
				N-acetylcysteine 2 g/day	TNF-α	
**McAnulty et al.** [93]	Healthy trained male adults	a 1-h run at a 3% grade and at ~80% VO_2_max	double-blind crossover (*n* = 14)	Resveratrol and quercetin (RQ)	ORAC	positive effect on reducing lipid peroxidation No positive effect on altering plasma antioxidant status and inflammation
				450 mg quercetin and 240 mg resveratrol on day 7 just prior to exercise and 225 mg quercetin and 120 mg resveratrol for 6 days prior to day 7	FRAP	
					TEAC	
					protein carbonyls	
					F_2_-isoprostanes	
					CRP	
					IL-8	
**Bailey et al.** [94]	healthy young men	90 min of intermittent shuttle-running	double-blind crossover (*n* = 38)	Combination of antioxidant supplements (vitamin C and E) 400 mg vitamin C (ascorbic acid), 268 mg vitamin E (RRR-α-tocopherol), 2 mg vitamin B6 (pyridoxine hydrochloride), 200 µg vitamin B9 (folic acid), 5 µg zinc sulphate monohydrate, and 1 µg vitamin B12 (cyanocobalamin)	Vitamin C	No positive effect on reducing inflammation and OS
					Vitamin E	
					F2-isoprostanes	
					Cortisol	
					Interleukin-6	
**Nieman et al.** [98]	ultramarathon athletes	160-km Western States Endurance Run (WSER)	double-blind crossover (*n* = 39)	Placebo or quercetin (1000 mg day^−1^)	proinflammatory and anti-inflammatory plasma cytokines	Positive effect on enhancing quercetin plasma levels muscle damage and No positive effect on reducing inflammatory cytokines
					cortisol	
					serum C-reactive protein (CRP)	
					creatine kinase (CK)	
**Tongtako et al.** [99]	patients with allergic rhinitis	walking and/or running on a treadmill at 65–70% heart rate reserve	(*n* = 27)	vitamin C 2 times/day (one pill of 1000 mg in the morning and one in the evening)	IL-2	Positive effect on reducing cytokine profiles
					IL-4	
**Díaz-Castro et al.** [100]	amateur athletes	run to Sierra Nevada from the city of Granada	-	Placebo or Phlebodium group (PG) Five capsules of 400 mg (250 mg of leaf extract and 150 mg of rhizome extract)	8-hydroxy-2-deoxyguanosine (8-OHdG)	Created a strong defense against inflammatory cytokines such as TNF-α
					Isoprostane	
					TNF-α	
					IL-6	
					IL-1ra	
**Díaz-Castro et al.** [102]	male amateur athletes	constant run (50 km) that combined several degrees of high effort (mountain run and ultraendurance)	-	Placebo or CoQ10 group	TNF-a	Positive effect on reducing muscle damage, oxidative stress and inflammatory cytokines
					IL-6	
					8-OHdG	
					Isoprostane	
**Ochoa et al.** [103]	amateur athletes	Run 50 km with almost 2800 m of ramp	-	melatonin-treated men (MG) and placebo-treated individuals (controls group, CG) (Five capsules of 3 mg)	TNF-αIL-6	Positive effect on reducing muscle damage, oxidative stress and inflammatory cytokines
					IL-1ra	
					8-OHdG	
					Isoprostane	
**Clifford et al.** [111]	recreationally active males	100-drop jumps	a double blind, independent groups design (*n* = 30)	high dose of beetroot juice (H-BT; 250 mL), low dose of beetroot juice (L-BT; 125 mL), or an isocaloric placebo (PLA; 250 mL)	countermovement jumps (CMJ)	Positive effect on reducing muscle pain Positive effect on enhancing CMJ
					pressure pain threshold	
					CK	
					IL-6	
					IL-8	
					TNF-α	
**Koenig et al.** [112]	Young women	downhill running (DR) on a treadmill at −9% grade	double-blind crossover (*n* = 16)	oat flour providing 9.2 mg AVA (AVA) or 0.4 mg AVA (Control, C)	TNF-α	Positive effect on reducing inflammatory cytokines
					neutrophil respiratory burst	
					CK	
					NF-κB	
					resting plasma GSH	

**Table 3 antioxidants-09-00886-t003:** The effect of different doses of antioxidant supplements on oxidative stress (OS).

Reference	Subjects	Exercise	Design	Supplement	Parameters	Results
**Chen et al.** [113]	Wistar rats	Exhaustive swimming	(*n* = 40)	Radix Pseudostellariae polysaccharides (RPPs) antioxidant supplement in different doses (100, 200, and 400 mg/kg body weight)	Hemoglobin	high-dose antioxidant supplementation was more effective in improving levels of SOD, GPx, and CAT further reduction in OS indicators
					blood lactate	
					Antioxidant enzymes (SOD, CAT)	
					GPx	
					MDA	
**Ren et al.** [114]	healthy mice	swimming exercise	(*n* = 90)	grass carp protein or peptide with low (1 mg/g.d) and high (5 mg/g.d)	liver glycogen	Positive effect on enhancing levels of SOD, GPX, and CAT
					gastrocnemius muscle glycogen	
					plasma glucose	
					serum lactic acid	
					blood urea nitrogen	
					SOD	
					GPx	
					CAT	
**Zheng et al.** [116]	pathogen free (SPF) Sprague-Dawley (SD) rats	incremental swimming exercise	(*n* = 60)	grass carp protein or peptide with low (1 mg/g.d) and high (5 mg/(g.d). low (20 mg kg^−1^ d^−1^) and high dose (100 mg kg^−1^ d^−1^) wheat peptide.	Exhaustive time	taking both doses of the supplement could boost GPx antioxidant levels. fatigue was delayed by taking a high-dose supplement over a lower dose
					MDA	
					Secretory	
					immunoglobulin A	
					5-hydroxytryptamine (5-HT)	
					SOD	
					GPx	
					Acetylcholinesterase	
					Caspase-3	
					IL-6	
					IL-8	
**Liu et al.** [117]	Sprague-Dawley rats	Strenuous exercise	experimental animal model	lycopene (a relatively high dose (7·8 mg/kg per d) and a relatively low dose (2·6 mg/kg per d))	Xanthine oxidase (XO)	No significant difference between the use of different antioxidant doses in the enhancement of Glutathione antioxidant enzymes
					myeloperoxidase (MPO)	
					MDA	
					GSH	
**Ranchordas et al.** [118]	Male and female sedentary-moderate trained	DOMS-inducing exercise	50 studies (*n* = 1089)	Antioxidant doses higher than Recommended Dietary Allowances (RDA)	Wide range of antioxidants, placebo-controlled	Exercise-induced muscle pain did not decrease even with high-dose antioxidant supplements

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
