# Peer review of "A Short Overview of Changes in Inflammatory Cytokines and Oxidative Stress in Response to Physical Activity and Antioxidant Supplementation"

_antioxidants, 2020, doi:10.3390/antiox9090886_

Round 1

Reviewer 1 Report

The manuscript by Taherkhani summarizes the effects of antioxidants and exercise on the production of reactive oxidative species and inflammatory cytokines. Overall, the review is well written and structured, especially the presented tables provide an interesting overview over the published work. My only major criticism concerns chapter 4. The discussed signaling pathways (NF-kB, calcineurin/NFAT, HSP) are not well integrated to the article. It is not clear how ROS/antioxidants influence the pathways and how this works on the molecular levels (e.g. by cysteine oxidation of the IKK complex etc.). In my opinion, some statements are misleading. In lane 177 (page 4) it is stated that “NFAT (transcriptional agents) has two members, including …..”. Maybe it would be better to clarify that the NFAT family consist of 4 calcium activated members (NFATc1 – NFATc4) and one form which senses osmotic stress (NFAT5). Also, some sentences in chapter 4 are difficult to understand such as “The brain is the content of proteins attached to a soluble calmodulin called calcineurin” (see lane 169, page 4). If the authors improve chapter 4 and integrate it better into the rest of the article, I may recommend the publications of the manuscript.

Author Response

Thank you for providing valuable and vital feedback on our manuscript. Your comments are highlighted in yellow in the original text. I hope that your expectations are met.

Reviewer 2 Report

The manuscript was prepared correctly, I have no comments.

The study is significant due to problems with the Reactive Oxygen Species. On the one hand, excess and accumulation ROS can cause oxidative damage, but on the other hand, ROS are essential in the proper functioning of many processes in the body.

The authors collected and reviewed original studies that have investigated changes in inflammatory cytokines and oxidative stress in response to exercise and antioxidant supplementation.

The importance of cytokines and oxidative changes during exercise, as well as the oxidation-reduction control of cytokine composition, are discussed in detail and correctly. The chapter on the effect of antioxidant supplements on inflammatory cytokines in the most detail was presented.

However, it would be worthwhile to focus on a more detailed discussion and explanation of some mechanisms in future studies. In addition, the manuscript, despite being well prepared, is not a very significant contribution to this field of science. Therefore, the Authors' next step should be to explore the topic further and conduct their research taking into account other factors such as gender, greater or even health.

Author Response

Thank you for your insightful and supportive comments. We take account of your concerns as well as those of another reviewer. In the near future, we are going to publish some prospective explanation on this topic considering other factors.

Reviewer 3 Report

Major point-

A substantial body of research suggests that exercise is a robust Zeitgeber of skeletal muscle clocks. As reviewed by authors, changes in inflammatory cytokines and oxidative stress in response to exercise are modulated by the type of exercise and intensity of training. Interestingly, there is increasing evidence implicates that both inflammatory cytokines (anti/pro) and oxidative stress are subjected to diurnal changes. Exercise can reset the molecular clocks of skeletal muscle and thus modulates inflammatory cytokine secretion and oxidative stress. The missing gap in the current review is the interplays of exercise and circadian rhythms on inflammatory cytokines and oxidative stress.

Minor point-

Exercise modifies gut microbiota, both in terms of abundance and diversity, which display diurnal changes. A brief discussion on how exercise alters gut microbiome in conjunction with antioxidant supplementation will be informative in this topic.

Author Response

We would like to thank you for useful comments. We mainly focused on the modulatory effect of antioxidant supplements on improving exercise-induced oxidative stress and inflammatory cytokines. Regarding the gap about interplay between exercise-induced inflammatory cytokines and oxidative stress, we discussed this topic in another of our publications.  Please refer to the following link.

https://www.mdpi.com/2076-3921/9/5/401

There are several research articles on the effect of exercise on the gut microbiome, and undoubtedly this is also a demanding topic. We had presumed that a brief discussion of such an interesting topic would not be sufficient, and therefore we mainly focused on the muscle metabolism. We plan to discuss this topic in greater detail in our next article soon.

Round 2

Reviewer 1 Report

Unfortunately, the requested major changes of chapter 4 regarding both precise scientific phrasing and clarification of misleading statements have been incompletely provided by the authors.

Statements such as "p50/p52 hemodimers" (lane 157), "NF-kB can be locked into cells...." (lane 160), "these proteins hide the nucleus binding sites" (lane 153-154), "Typical proteins have a great effect on removing the bond between DNA molecules" (lane 152-153) or "...is one of the key regulators of the gene expression within intracellular calcium concentration..." (lane 186-187) are either misleading or scientifically incorrect.

Therefore I cannot recommend the publication of this review article.

Author Response

Dear reviewer;

Thank you for your insightful and helpful comments. Our responses to your comments are highlighted in green in the text of the re-submitted MS. We rewrote the part that you thought needed revision along with adding new content. We hope that your expectations are met.